# Diet and BMI Correlate with Metabolite Patterns Associated with Aggressive Prostate Cancer

**DOI:** 10.3390/nu14163306

**Published:** 2022-08-12

**Authors:** Zoe S. Grenville, Urwah Noor, Mathilde His, Vivian Viallon, Sabina Rinaldi, Elom K. Aglago, Pilar Amiano, Louise Brunkwall, María Dolores Chirlaque, Isabel Drake, Fabian Eichelmann, Heinz Freisling, Sara Grioni, Alicia K. Heath, Rudolf Kaaks, Verena Katzke, Ana-Lucia Mayén-Chacon, Lorenzo Milani, Conchi Moreno-Iribas, Valeria Pala, Anja Olsen, Maria-Jose Sánchez, Matthias B. Schulze, Anne Tjønneland, Konstantinos K. Tsilidis, Elisabete Weiderpass, Anna Winkvist, Raul Zamora-Ros, Timothy J. Key, Karl Smith-Byrne, Ruth C. Travis, Julie A. Schmidt

**Affiliations:** 1Cancer Epidemiology Unit, Nuffield Department of Population Health, University of Oxford, Oxford OX3 7LF, UK; 2Nutrition and Metabolism Branch, International Agency for Research on Cancer, World Health Organization, 69008 Lyon, France; 3Department of Epidemiology and Biostatistics, School of Public Health, Imperial College London, London W2 1PG, UK; 4Ministry of Health of the Basque Government, Sub Directorate for Public Health and Addictions of Gipuzkoa, 20013 San Sebastian, Spain; 5Biodonostia Health Research Institute, Epidemiology of Chronic and Communicable Diseases Group, 20014 San Sebastián, Spain; 6Spanish Consortium for Research on Epidemiology and Public Health (CIBERESP), Instituto de Salud Carlos III, 28029 Madrid, Spain; 7Department of Clinical Sciences, Lund University, 221 84 Malmö, Sweden; 8CIBER of Epidemiology and Public Health (CIBERESP), 28029 Madrid, Spain; 9Department of Epidemiology, Regional Health Council, IMIB-Arrixaca, Murcia University, 30008 Murcia, Spain; 10Skåne University Hospital, 214 28 Malmö, Sweden; 11Department of Molecular Epidemiology, German Institute of Human Nutrition, 14558 Nuthetal, Germany; 12Epidemiology and Prevention Unit, Fondazione IRCCS Istituto Nazionale dei Tumori, 20133 Milan, Italy; 13Department of Cancer Epidemiology, German Cancer Research Center (DKFZ), 69120 Heidelberg, Germany; 14Cancer Epidemiology Unit, Department of Medical Sciences, University of Turin, 10124 Turin, Italy; 15Navarra Public Health Institute, 31003 Pamplona, Spain; 16Navarra Institute for Health Research (IdiSNA), 31008 Pamplona, Spain; 17Danish Cancer Society Research Center, DK-2100 Copenhagen, Denmark; 18Department of Public Health, Aarhus University, DK-8000 Aarhus, Denmark; 19Escuela Andaluza de Salud Pública (EASP), 18011 Granada, Spain; 20Instituto de Investigación Biosanitaria ibs.GRANADA, 18012 Granada, Spain; 21Centro de Investigación Biomédica en Red de Epidemiología y Salud Pública (CIBERESP), 28029 Madrid, Spain; 22Department of Preventive Medicine and Public Health, University of Granada, 18071 Granada, Spain; 23Department of Public Health, University of Copenhagen, DK-1353 Copenhagen, Denmark; 24Department of Hygiene and Epidemiology, School of Medicine, University of Ioannina, 45110 Ioannina, Greece; 25International Agency for Research on Cancer, World Health Organization, 69008 Lyon, France; 26Sustainable Health, Department of Public Health and Clinical Medicine, Umeå University, 901 87 Umeå, Sweden; 27Department of Internal Medicine and Clinical Nutrition, The Sahlgrenska Academy, University of Gothenburg, 413 45 Gothenburg, Sweden; 28Unit of Nutrition and Cancer, Cancer Epidemiology Research Programme, Catalan Institute of Oncology (ICO), Bellvitge Biomedical Research Institute (IDIBELL), L’Hospitalet de Llobregat, 08908 Barcelona, Spain; 29Department of Clinical Epidemiology, Department of Clinical Medicine, University Hospital, Aarhus University and Aarhus, DK-8200 Aarhus N, Denmark

**Keywords:** metabolites, diet, prostate cancer, cross-sectional

## Abstract

Three metabolite patterns have previously shown prospective inverse associations with the risk of aggressive prostate cancer within the European Prospective Investigation into Cancer and Nutrition (EPIC). Here, we investigated dietary and lifestyle correlates of these three prostate cancer-related metabolite patterns, which included: 64 phosphatidylcholines and three hydroxysphingomyelins (Pattern 1), acylcarnitines C18:1 and C18:2, glutamate, ornithine, and taurine (Pattern 2), and 8 lysophosphatidylcholines (Pattern 3). In a two-stage cross-sectional discovery (*n* = 2524) and validation (*n* = 518) design containing 3042 men free of cancer in EPIC, we estimated the associations of 24 dietary and lifestyle variables with each pattern and the contributing individual metabolites. Associations statistically significant after both correction for multiple testing (False Discovery Rate = 0.05) in the discovery set and at *p* < 0.05 in the validation set were considered robust. Intakes of alcohol, total fish products, and its subsets total fish and lean fish were positively associated with Pattern 1. Body mass index (BMI) was positively associated with Pattern 2, which appeared to be driven by a strong positive BMI-glutamate association. Finally, both BMI and fatty fish were inversely associated with Pattern 3. In conclusion, these results indicate associations of fish and its subtypes, alcohol, and BMI with metabolite patterns that are inversely associated with risk of aggressive prostate cancer.

## 1. Introduction

Metabolomics is a rapidly evolving field, which involves the measurement of multiple metabolites with an aim of establishing biomarkers of exposure and disease risk [1,2]. Several observational studies have measured prediagnostic blood metabolites in order to identify novel risk factors and pathways in prostate cancer aetiology [1,2,3,4], including analyses of metabolite profiles, as well as specific analytes. In a previous case-control study nested within the European Prospective Investigation into Cancer and Nutrition (EPIC) cohort, three patterns (treelet components) were identified and assessed in relation to risk for prostate cancer [1]; a metabolite pattern of 64 diacyl- and acyl-alkyl phosphatidylcholines and three hydroxysphingomyelins, as well as a metabolite pattern of two acylcarntines, glutamate, ornithine, and taurine, were both found to be inversely associated with risk of advanced and aggressive prostate cancer. Furthermore, a metabolite pattern of eight lysophosphatidylcholines was also observed to be inversely associated with risk of advanced prostate cancer and prostate cancer death [1]. Data from other cohorts have also supported inverse associations of glycerophospholipids [3] and acylcarnitine C18:2 [4] with risk of aggressive prostate cancer. 

Blood metabolite concentrations are affected by both internal and external factors, including modifiable factors, such as diet and body mass index (BMI) [5,6]. Thus, a better understanding of how these factors are associated with prostate cancer-related metabolite patterns might offer insights into possible avenues for prostate cancer prevention. 

This cross-sectional study nested in the EPIC cohort aimed to investigate associations of dietary variables and BMI with metabolite patterns previously found to be inversely associated with more aggressive prostate cancer subtypes. 

## 2. Materials and Methods

### 2.1. Study Population

EPIC is a multi-center prospective cohort study, which recruited approximately 500,000 Europeans, including 153,457 men, between 1992 and 2000. The current analyses include men mainly aged between the ages of 35 and 70 years at recruitment from 19 centers in seven countries (Denmark, Germany, Italy, Netherlands, Spain, Sweden and United Kingdom). 139,600 of the men provided a blood sample. All participants in the EPIC study provided written informed consent, and the study was approved by the ethics committees of the International Agency for Research on Cancer (IARC) and all participating centers [7].

Men were eligible for the current study if they (1) were free of cancer (except non-melanoma skin cancer) at baseline; (2) had a known date of blood collection; (3) had been included as control participants in one of four case-control studies on metabolite concentrations and cancer risk nested within the EPIC cohort (on prostate [1], colorectal [8], kidney [9], and liver cancer [10]), hereafter referred to as sub-studies, with available blood concentrations of all of the metabolites included in the metabolite patterns that were previously found to be associated with more aggressive prostate cancer subtypes; and (4) had blood samples that were included in an analytical batch that had at least 10 samples, to ensure proper normalization of metabolite concentrations. Thus, data for 3198 men were available for this study.

### 2.2. Laboratory Measurements

For participants from Germany, Italy, the Netherlands, Spain and the UK, biological samples are stored at IARC in plastic straws at −196 °C (details published elsewhere) [7]. In Sweden and Denmark, blood samples are stored in tubes in local repositories; in Sweden, the samples are kept in freezers at −80 °C, and in Denmark in nitrogen vapor at −150 °C [7]. 

Regardless of sub-study, all samples were previously assayed at IARC in Lyon, France using the AbsoluteIDQ^®^ p180 Kit (Biocrates Life Sciences AG, Innsbruck, Austria), and following the procedure recommended by the vendor. To quantify metabolites, liquid chromatography mass spectrometry (LC-MS) was applied. All samples were assayed using one LC instrument (Agilent 1290, Santa Clara, CA, USA) coupled with one of two different triple quadrupole MS instruments (Triple Quad 4500, AB Sciex, Framingham, MA, USA for prostate and colorectal cancer [1,8] and Q-Trap 5500, AB Sciex, MA for liver and kidney cancer [10,11]; Appendix A). Of note, within each sub-study a single pair of LC-MS instruments was used for all samples [12]. 118 common metabolites were measured across all sub-studies [12].

Metabolite values outside the measurable range, including metabolite values below the batch-specific limit of detection (LOD), below the kit-specific lower limit of quantification (LLOQ), and above the kit-specific upper limit of quantification (ULOQ), were imputed to LOD/2, LLOQ/2, and ULOQ, respectively. 

### 2.3. Diet, BMI, and Covariate Data

Detailed information on dietary, lifestyle, and anthropometric data was gathered at recruitment, previously described in Riboli et al. [7]. In order to determine usual dietary intakes, center- or country-specific validated dietary questionnaires covering the previous 12 months were used [13]. The dietary variables (continuous, consumption in g/day) investigated in this study were intakes of total dairy (sum of milk, cheese, and yogurt), milk, cheese, yogurt, eggs, red meat, poultry, processed meat, total fish products (refers to fish and shellfish combined), total fish (subset of total fish products), fatty fish (subcategory of total fish), lean fish (subcategory of total fish), fats and oils (sum of butter, margarine, and vegetable oils), butter, margarine, vegetable oils, total vegetables (sum of leafy, root, and fruiting vegetables), leafy vegetables, root vegetables, fruiting vegetables, total fruit, cereals and cereal products, and alcohol. In order to reflect average daily consumption, increments were chosen for each dietary variable to represent typical intakes in an average European male population (Table 1).

BMI (continuous, kg/m^2^) was also examined as a possible correlate of the metabolite patterns, calculated from weight and height (measured, except self-reported in some participants in the EPIC-Oxford cohort) [8].

### 2.4. Statistical Analysis

#### 2.4.1. Participant Characteristics

Participants’ characteristics at baseline were summarized using frequencies for categorical variables and mean (standard deviation) for continuous variables.

#### 2.4.2. Normalization of Metabolite Concentrations

A statistical pipeline has been developed for the EPIC metabolomics data [12] and was applied in this analysis to the raw metabolite concentrations. Metabolites with more than 25% missing values in each study were removed. For the remaining missing data, if no more than 50% were missing in the batch, values were imputed to the batch-specific median (of the considered metabolite); if more than 50% were missing in the batch, they were otherwise imputed to the median of the medians of the measured values in the other batches. Log-transformed concentrations of the metabolites were then normalized using linear mixed-effects models to remove unwanted variations due to study, batch, and center; study and batch were included as random effects and center was included as a fixed effect in the models. Corrected metabolite concentrations analyzed in this work correspond to residuals from the individual models. This pipeline was shown to be efficient in removing unwanted variability and improving the comparability of measurements acquired across the different cancer-specific studies [12]. 

#### 2.4.3. Metabolite Patterns

Patterns in metabolite profiles were previously identified using treelet transform in an EPIC nested case-control study of prostate cancer [1]. In summary, treelet transform is a linear dimension-reduction method aiming at summarizing the metabolite variables into fewer latent variables that best capture the observed variation in the overall set of metabolites [20,21]. Schmidt et al. identified three treelet components (henceforth referred to in the text as metabolite patterns, which together explained 31.4% of the total variance in metabolite concentrations), all of which were found to have an inverse association with advanced and/or aggressive prostate cancer risk. The first metabolite pattern (Pattern 1) had positive loadings on diacyl-phosphatidylcholines (PC aa; *n* = 31) and acyl-alkyl-phosphatidylcholines (PC ae; *n* = 33), as well as three hydroxysphingomyelins (SM(OH)): C14:1, C16:1, and C22:2. The second metabolite pattern (Pattern 2) had positive loadings on acylcarnitines C18:1 and C18:2, and the amino acids glutamate, ornithine, and taurine. Finally, the third metabolite pattern (Pattern 3) had positive loadings on eight lysophosphatidylcholines (lyso PC a): C16:0, C16:1, C17:0, C18:0, C18:1, C18:2, C20:3, and C20:4 (Table 2) [1]. Each metabolite pattern was scaled to units of one standard deviation (SD), as done in the previous study [1].

#### 2.4.4. Correlates of Metabolites

After excluding participants with missing values for time at blood collection (78), fasting status (65), energy intake (2), BMI (23), and level of education (31), the current cross-sectional analysis included data from 3042 participants. These data were subsequently split into a discovery set (*n* = 2524; 83% of the population) and a validation set (*n* = 518; 17.0% of the population). Specifically, the discovery set included controls from the prostate cancer sub-study that were used in the identification of metabolite patterns, while the validation set comprised controls from the other three sub-studies (kidney [9], liver [10], and colorectal [8] cancer). A discovery-validation set design was chosen to both reduce the in-sample bias from the samples used to determine patterns, and to afford an external validation for any associations that appeared statistically significant in initial analyses. 

First, analyses were run in the discovery set. For each of the three metabolite patterns and each dietary or lifestyle variable, a linear regression model was run with the metabolite pattern as the dependent variable. Models were adjusted for age at blood collection (continuous), time of day of blood collection (continuous), fasting status at blood collection (<3 h since last meal, 3–6 h, >6 h, and missing), baseline education level (primary/no schooling, secondary, professional/technical, university/higher, not specified, and missing), physical activity (Cambridge index [22]: inactive, moderately inactive, moderately active, active, and missing), smoking status (never, former, current, and missing), energy (continuous, kcal/day) and alcohol intakes (continuous, g/day), and BMI (continuous, kg/m^2^). Models that examined alcohol intake and BMI as main exposures were not adjusted for alcohol intake and BMI, respectively.

In the discovery set, to account for multiple testing, we used a Benjamini-Hochberg false discovery rate (FDR) by metabolite pattern at a 5% threshold to define statistical significance [23]. Each statistically significant association in the discovery set was re-assessed in the validation set, using the same variables and adjusted models. Results from the analyses in the validation set were not corrected for multiple testing. Associations between exposures and metabolites that passed the FDR threshold in the discovery set, and the significance threshold in the validation set (*p* < 0.05), were considered robust.

#### 2.4.5. Individual Metabolite Analysis

A supplementary analysis was conducted of dietary exposures and BMI with the individual metabolites that contributed to metabolite patterns with which they were robustly associated (Appendix A). Models were adjusted as described above for the main analysis. Individual metabolite values were log-transformed. Linear regressions were run in the overall dataset (*n* between 2136 and 3042, depending on exposure). To account for multiple testing, dietary and lifestyle correlates of metabolites that passed the FDR of 0.05 were determined to be statistically significant.

## 3. Results

### 3.1. Participant Characteristics

Main characteristics of the participants, overall and in the discovery and validation sets, are shown in Table 3. 46.4% of men in the discovery set and 31.7% of men in the validation set were not considered fasting at blood collection (<3 h since last meal), while 32.0% and 45.7% of men in the discovery and validation sets, respectively, were fasting (>6 h since last meal). Otherwise, participant characteristics were relatively similar in the discovery and validation sets.

### 3.2. Correlates of Metabolite Patterns

Figure 1 depicts the betas and 95% confidence intervals for associations between the metabolite patterns and selected potential correlates in the discovery and validation sets. Appendix A shows the full results for betas, *p*-values, and P_adj_ values (*p*-values after adjusting for multiple testing in the discovery set) for the exposure–metabolite pattern associations in the discovery and validation sets. Associations with individual metabolites are shown in Appendix A. 

Of the nine exposures that passed the significance threshold after adjusting for multiple testing in the discovery set, intakes of total fish products, total fish, lean fish, and alcohol all remained statistically significantly (*p* < 0.05) positively associated with pattern 1 in the validation set (Figure 1a). In the analysis of individual metabolites contributing to pattern 1, the associations with the lowest *p*-values included alcohol with PC aa C32:1, C34:1, and C36:4, and total fish products, total fish, and lean fish with PC aa C42:2 (see Appendix A for full results).

For metabolite pattern 2, seven exposures passed the significance threshold after multiple testing in the discovery set, though only BMI remained positively and statistically significantly associated with the metabolite pattern in the validation set (Figure 1b). This relationship appeared to be strongly driven by a strong, positive BMI-glutamate association (Appendix A).

Of the six exposures that were significant after multiple testing in the discovery set, fatty fish intake and BMI remained statistically significantly inversely associated with pattern 3 in the validation set (Figure 1c). In the analysis of individual metabolites, BMI was strongly and inversely associated with all eight lyso PCs loading on the metabolite pattern, while fatty fish was significantly inversely associated with lyso PCs C16:1, C18:0, C18:1, C20:3, and C20:4 (Appendix A).

## 4. Discussion

This large cross-sectional study identified several dietary factors and BMI as correlates of metabolite patterns that have previously been shown to associate inversely with more aggressive prostate cancer subtypes. The intakes of alcohol, total fish products, and its subsets total fish and lean fish, were all positively associated with a metabolite pattern with higher concentrations of 31 PC aas, 33 PC aes, and three hydroxysphingomyelins. BMI was positively associated with the second metabolite pattern of two acylcarnitines, glutamate, ornithine, and taurine, and an individual metabolite analysis showed that this was driven by a specific association with glutamate. Finally, BMI and fatty fish intake were inversely associated with scores of a third metabolite pattern of eight lyso PCs as no additional associations of dietary variables or BMI with metabolite patterns were validated.

Comparing these results to previous studies is complex; this analysis primarily investigated metabolite patterns rather than individual metabolites. Broadly in line with our results, however, positive associations of alcohol intake with some of the metabolites loading on metabolite pattern 1 (phosphatidylcholines and hydroxysphingomyelins) have previously been reported in other analyses [24,25,26,27].

Though there are limited prior studies of associations between fish intake and metabolites, a positive association between fish intake and certain phosphatidylcholines has also been previously reported in an analysis in the TwinsUK cohort [24], and in an intervention study at the University of Otago [28]. This may be partially attributed to fish being a dietary source of choline, which is a requirement for hepatic phosphatidylcholine biosynthesis [29].

The positive association between BMI and the metabolites loading on metabolite pattern 2 (driven largely by glutamate) was consistent with findings in other studies, including the Framingham Heart Study, and the Malmö Diet and Cancer Study [30]. Additionally, a Mendelian randomization analysis suggested that the positive effect of BMI on circulating glutamate may be causal [9]. Previous studies have also demonstrated that glutamate is positively linked to visceral obesity [8,9,30,31].

This study found inverse associations of fatty fish intake and BMI with pattern 3, which comprised eight lyso PCs. To date, there are still limited data available regarding the effects of diet on lyso PC concentrations. However, an 8-week sequential therapy clinical trial for adults with diabetes mellitus found a reduction in circulating levels of lyso PC C16:1 after consistent fish oil supplementation [32], which may support the current study’s findings that fatty fish is inversely associated with a metabolite pattern of eight lyso PCs, including lyso PC C16:1.

For BMI and metabolite pattern 3, similar to our study, the aforementioned Mendelian randomization analysis on BMI and metabolites also found an inverse effect of BMI on blood levels of lyso PCs C18:1 and C18:2 [9], both of which were included in the pattern.

The current analyses have identified dietary (fish and alcohol) and anthropometric (BMI) correlates of three metabolite patterns that were previously found to be inversely associated with more aggressive prostate cancer subtypes [1]. The implications of these associations are not yet clear; fish [33,34,35,36,37,38,39,40,41] and alcohol intakes [42,43,44,45,46,47,48,49,50,51] are not established risk factors for prostate cancer, while any associations of BMI with prostate cancer risk, of which positive associations with advanced disease and death have previously been reported [52], might be due to differences in the timing of detection of prostate cancer in men with obesity compared to men with a normal BMI [27,53,54,55,56,57]. Furthermore, research is ongoing to determine whether the metabolite pattern–prostate cancer associations previously reported are likely to be causal.

## 5. Strengths and Limitations

A major strength of this study is its large sample size owing to the pooling of metabolomics data from four sub-studies within EPIC. Furthermore, using the metabolite patterns as outcome variables accounted for correlations between metabolites [1]. In addition, having access to a wide variety of collected exposure data allowed for the investigation of a range of potential dietary variables and BMI, and adjustment for potential confounding factors. Finally, the discovery/validation approach likely reduced the in-sample bias due to participant overlap between those used to derive metabolite patterns and those used to validate the diet- and BMI-metabolite pattern associations.

One limitation of this study is its cross-sectional design, which prevents drawing any definitive conclusions about the temporality and causality of the reported associations. Potential heterogeneity in metabolite concentrations, such as between the four sub-studies, was addressed by applying a dedicated pipeline to the data prior to statistical analyses [12], and the analytical protocol used has demonstrated high reproducibility between instruments [57]. To assess dietary intakes, food frequency questionnaires were used in most EPIC centers, which can result in some measurement error due to the misreporting of food consumption, recall bias, or errors related to the food composition tables used. Despite this, numerous pilot and cross-sectional studies have supported the reproducibility and validity of the food frequency questionnaire method [27,58]. Finally, this study was conducted on a European population, and while there is limited information on racial and ethnic diversity of the participants, it is expected that the participants are primarily of European ancestry. This limited diversity may hamper the generalizability of the current findings to non-European populations. Future research should evaluate associations in different ethnic and racial groups to provide a more generalizable understanding of determinants of the circulating metabolome.

## 6. Conclusions

This large, cross-sectional study in European men indicates that BMI and intakes of total fish products, fish subtypes, and alcohol are associated with the metabolite patterns that have been previously linked to a lower risk of aggressive prostate cancer tumor subtypes. The nature and possible causality of these associations warrants further investigation.

## Figures and Tables

**Figure 1 nutrients-14-03306-f001:**
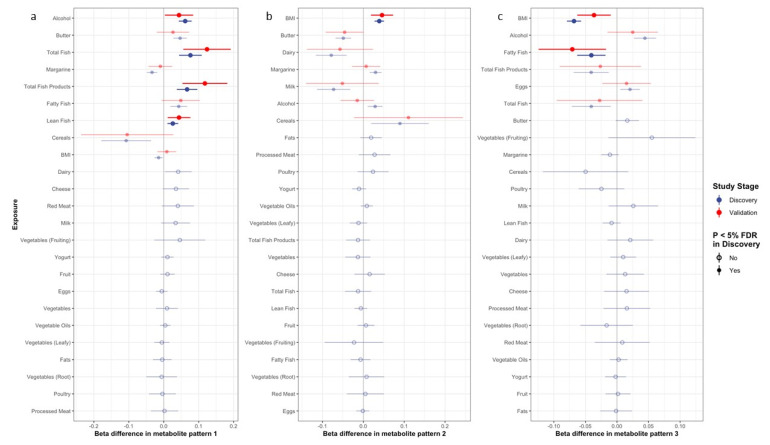
Association of dietary and lifestyle factors with metabolite pattern 1 (**a**), pattern 2 (**b**), and pattern 3 (**c**). Estimates presented in bold are for dietary and lifestyle factors with both significant P_adj_ associations in the discovery dataset (blue) and *p* < 0.05 in the validation dataset (red). Estimates with associations that do not pass FDR threshold are presented with hollow circles.

**Table 1 nutrients-14-03306-t001:** Increments for each dietary variable.

Dietary Variable	Increment (Grams per Day)
Dairy	200 g [14]
Milk	200 g [14]
Cheese	30 g [15]
Yogurt	30 g [15]
Eggs	7 g [16]
Total fish products	30 g [17]
Total fish	30 g [17]
Lean fish	10 g [17]
Fatty fish	10 g [17]
Red meat	40 g [16]
Poultry	20 g [16]
Processed meat	40 g [16]
Fats and oils	10 g [16]
Butter	5 g [16]
Margarine	5 g [16]
Vegetable oils	5 g [18]
Fruits	100 g [19]
Vegetables	100 g [19]
Leafy vegetables	25 g [19]
Root vegetables	25 g [19]
Fruiting Vegetables	100 g [19]
Cereals and cereal products	200 g [14]
Alcohol	10 g [14]

**Table 2 nutrients-14-03306-t002:** Metabolite Patterns and their loadings.

Metabolite Pattern	Contributing Metabolites All with Positive Loadings	Percent Explained Variance (%)
1	64 diacyl and acyl-alkyl phosphatidylcholines; (SM (OH) C14:1, SM (OH) C16:1, and SM (OH) C22:2)	21.5
2	Acylcarnitines C18:1 and C18:2, glutamate, ornithine, and taurine	5.2
3	Lyso PC a C16:0, lyso PC a C16:1, lyso PC a C17:0, lyso PC a C18:0, lyso PC a C18:1, lyso PC a C18:2, lyso PC a C20:3, lyso PC a C20:4	4.7

**Table 3 nutrients-14-03306-t003:** Main characteristics of men included in the analysis, overall and separately in discovery and validation sets.

Participant Characteristics	Overall (*n* = 3198)	Discovery (*n* = 2640)	Validation (*n* = 558)
Age at blood collection (years)	57.2 (7.2)	57.5 (7.1)	56.0 (7.8)
Fasting status at blood collection (time since last meal) (*n* (%))			
<3 h	1402 (43.8)	1225 (46.4)	177 (31.7)
3–6 h	631 (19.7)	526 (19.9)	105 (18.8)
>6 h	1100 (34.4)	845 (32.0)	255 (45.7)
Missing	65 (2.0)	44 (1.7)	21 (3.8)
**Socio-economic and lifestyle factors (*n* (%))**			
Educational level			
Primary/no schooling	1216 (38.0)	992 (37.6)	224 (40.1)
Secondary	347 (10.9)	289 (11.0)	58 (10.4)
Technical/professional	744 (23.3)	612 (23.2)	132 (23.7)
University or higher	761 (23.8)	633 (24.0)	128 (22.9)
Not specified	99 (3.1)	88 (3.3)	11 (2.0)
Missing	31 (0.9)	26 (0.9)	5 (0.9)
Physical activity (Cambridge Index)			
Inactive	722 (22.6)	582 (22.1)	140 (25.1)
Moderately inactive	1048 (32.8)	869 (32.9)	179 (32.1)
Moderately active	731 (22.9)	609 (23.1)	122 (21.9)
Active	637 (19.9)	523 (19.8)	114 (20.4)
Missing	60 (1.9)	57 (2.2)	3 (0.5)
Smoking status			
Never	1025 (32.1)	843 (31.9)	182 (32.6)
Former	1374 (43.0)	1129 (42.8)	245 (43.9)
Current	765 (23.9)	640 (24.2)	125 (22.4)
Missing	34 (1.1)	28 (1.1)	6 (1.1)
Alcohol consumption at recruitment			
Non-drinker (<0.1 g/day)	286 (8.9)	235 (8.9)	51 (9.1)
>0.1–3 g/day	432 (13.5)	360 (13.6)	72 (12.9)
>3–12 g/day	730 (22.8)	605 (22.9)	125 (22.4)
>12–24 g/day	644 (20.1)	539 (20.4)	105 (18.8)
>24 g/day	1106 (34.6)	901 (34.1)	205 (36.7)
**Anthropometric variables (mean (SD))**			
Height (cm)	172.7 (7.0)	172.7 (7.1)	173.0 (6.7)
BMI (kg/m^2^)	26.9 (3.4)	26.9 (3.4)	26.9 (3.3)
**Dietary variables (g/day) (mean (SD))**			
Total energy (kcal/day)	2390 (649)	2375 (650)	2440(641)
Dairy	303 (229)	302 (227)	306 (237)
Milk	198 (205)	199 (204)	195 (212)
Cheese	34.5 (35.2)	33.6 (34.1)	38.7 (39.6)
Yogurt	38.9 (70.4)	37.5 (67.2)	45.8 (83.5)
Egg	18.6 (17.9)	18.4 (18.1)	19.5 (16.7)
Total fish products	40.9 (41.8)	40.9 (41.8)	41.0 (41.6)
Total fish	35.1 (38.3)	35.2 (38.0)	34.8 (39.4)
Lean fish	24.9 (31.8)	25.1 (31.6)	24.2 (33.0)
Fatty fish	12.8 (18.2)	12.8 (18.4)	13.0 (17.5)
Red meat	49.6 (36.6)	49.0 (36.2)	52.5 (38.2)
Processed meat	45.9 (42.7)	45.9 (43.4)	45.9 (38.8)
Poultry	21.9 (21.2)	21.9 (21.0)	21.8 (22.4)
Fats and oils	32.6 (17.4)	32.3 (17.3)	33.9 (17.6)
Butter	5.24 (10.5)	5.45(10.6)	4.26 (9.68)
Margarine	9.74 (14.7)	9.69 (14.5)	9.93 (15.8)
Vegetable oil	16.5 (17.7)	16.1 (17.5)	18.4 (18.5)
Vegetables	190 (129)	191 (130)	186 (128)
Leafy vegetables	30.4 (49.0)	30.0 (49.1)	32.2 (48.6)
Fruiting vegetables	67.6 (56.3)	67.0 (56.1)	70.2 (57.1)
Root vegetables	19.6 (24.2)	20.1 (24.6)	17.5 (22.0)
Fruit	236 (206)	233 (204)	251 (214)
Cereal	257 (139)	253 (134)	273 (161)
**Scores for metabolite patterns**			
Pattern 1 (geometric mean (SD))	10.2 (1.30)	10.2 (1.30)	10.2 (1.20)
Pattern 2 (geometric mean (SD))	1.98 (0.44)	1.98 (0.44)	1.98 (0.45)
Pattern 3 (geometric mean (SD))	6.13 (0.61)	6.13 (0.61)	6.13 (0.61)

BMI was missing for 23 (0.7%) participants (21 in discovery set). Total energy intake was missing for 2 (0.01%) participants (2 in discovery set). All dietary exposures (except fatty fish and lean fish) were missing for 2 (0.01%) participants (2 in discovery set). Fatty fish was missing for 426 (13%) participants (354 in discovery set). Lean fish was missing for 925 (29%) participants (780 in discovery set). Abbreviations: BMI, body mass index.

## Data Availability

EPIC data are available for investigators who seek to answer important questions on health and disease in the context of research projects that are consistent with the legal and ethical standard practices of IARC/WHO and the EPIC Centers. The primary responsibility for accessing the data belongs to the EPIC centers that provided them. For information on how to submit an application for gaining access to EPIC data and/or biospecimens, please follow the instructions at http://epic.iarc.fr/access/index.php.

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
