# Peer review of "Diet and BMI Correlate with Metabolite Patterns Associated with Aggressive Prostate Cancer"

_nutrients, 2022, doi:10.3390/nu14163306_

Round 1
Reviewer 1 Report
Grenville and colleagues conducted a study aimed at identifying dietary and lifestyle correlates of three metabolite patterns associated with aggressive prostate cancer. This is a very well written manuscript reporting findings from a statistically robust study with a large samples size and based on a discovery-validation approach. Metabolite values were corrected for batch effects and study site using a published pipeline. Results for dietary data were calculated for increments that represent typical intakes in an average European male population. Analyses were adjusted for a series of relevant factors and included only controls from previous case-control studies. I only have minor comments and suggestions.
Briefly describe QC/QA procedures for the metabolomics data. Were any metabolites excluded based on findings from QC/QA? How were missing metabolite values treated?
How do the authors interpret the different directions of associations of BMI with patterns 2 and 3, given that both patterns are inversely associated with risk?
I strongly suggest acknowledging the lack of racial/ethnic diversity in the study population which results in limited generalizability of the current findings.
Add a column with % explained variance to Table 2.
Given the complexity and length of Table 4, I suggest showing these results in a Figure, with three vertical panels, one for each pattern. Each panel can include all exposures (in the same order) and show effect estimates and 95%CI (horizontally), with discovery and validation results below each other. Exposure that belong together can be colored accordingly. Validated exposures can be marked accordingly. Table 4 can then be moved to the supplement.
I suggest using left alignment for all table columns.
Author Response
AUTHORS' RESPONSE TO REVIEWERS
We thank the reviewers for their detailed and helpful comments.
Below is a point-by-point response to the reviewers’ comments.
Reviewer #1:
Grenville and colleagues conducted a study aimed at identifying dietary and lifestyle correlates of three metabolite patterns associated with aggressive prostate cancer. This is a very well written manuscript reporting findings from a statistically robust study with a large samples size and based on a discovery-validation approach. Metabolite values were corrected for batch effects and study site using a published pipeline. Results for dietary data were calculated for increments that represent typical intakes in an average European male population. Analyses were adjusted for a series of relevant factors and included only controls from previous case-control studies. I only have minor comments and suggestions.
- Briefly describe QC/QA procedures for the metabolomics data. Were any metabolites excluded based on findings from QC/QA? How were missing metabolite values treated?
Authors’ response:
Thank you for your question. Detailed below is further information on the QC/QA procedures for the metabolomics data:
Filtering:
- Metabolites with more than 25% missing values in each study were removed.
Imputation:
< known LOD (limit of detection): imputed to batch-specific LOD/2
< known LLOQ (lower limit of quantification): imputed to LLOQ/2
> known ULOQ (upper limit of quantification): imputed to ULOQ
- Truly missing data were imputed to the batch-specific median (of the considered metabolite) if no more than 50% were missing in the batch; otherwise, missing data were imputed to the median of the medians of the measured values in the other batches.
Authors’ action:
We have added further information to the Methods section, please see lines 146-149 and 175-180:
“Metabolite values outside the measurable range, including metabolite values below the batch-specific limit of detection (LOD), below the kit-specific lower limit of quantification (LLOQ), and above the kit-specific upper limit of quantification (ULOQ), were imputed to LOD/2, LLOQ/2, and ULOQ, respectively.”
“Metabolites with more than 25% missing values in each study were removed. For the remaining missing data, if no more than 50% were missing in the batch, values were imputed to the batch-specific median (of the considered metabolite); if more than 50% were missing in the batch, they were otherwise imputed to the median of the medians of the measured values in the other batches.”
- How do the authors interpret the different directions of associations of BMI with patterns 2 and 3, given that both patterns are inversely associated with risk?
Authors’ response:
The manuscript focus is on the associations of the dietary variables and BMI with metabolite patterns and levels of individual metabolites, and without further research it is difficult to interpret the findings with respect to conclusions about possible prostate cancer risk factors, such as BMI, and any effects mediated via changes in metabolite levels. It is still unclear whether the associations observed (of dietary factors and BMI with metabolite levels, and of metabolite levels with prostate cancer risk) are causal; it may be, for example, that the metabolite-prostate cancer association is due to the effect of the tumor on metabolite levels and further analyses, such as observational analyses after extended follow-up in the prospective studies, are needed to assess this possible reverse causality. Furthermore, it is not clear that the magnitude of associations of the exposures with metabolites and of metabolite levels with risk are large enough for us to be able to detect an association of the exposures with risk acting via the metabolites.
- I strongly suggest acknowledging the lack of racial/ethnic diversity in the study population which results in limited generalizability of the current findings.
Authors’ response:
Thank you for your insight. We have now added a comment acknowledging the European-specific population in the limitations section.
Authors’ action:
We have added further information to the Strengths and Limitations Section. Please see page 10, lines 378-384:
“This study was conducted in a European population, and while there is limited information on racial and ethnic diversity of the participants, it is expected that the participants are primarily of European ancestry. This limited diversity may hamper the generalizability of the current findings to non-European populations. Future research should evaluate associations in different ethnic and racial groups to provide a more generalisable understanding of determinants of the circulating metabolome.”
- Add a column with % explained variance to Table 2.
Authors’ response: Thank you for your comment. We have now added a column showing the % explained variance in Table 2 (page 5, line 204).
Authors’ action:
Please see page 5, “Table 2”:
|
Metabolite Pattern |
Contributing metabolites all with positive loadings |
Percent explained variance (%) |
|
1 |
64 diacyl and acyl‐alkyl phosphatidylcholines; (SM (OH) C14:1, SM (OH) C16:1, and SM (OH) C22:2) |
21.5 |
|
2 |
Acylcarnitines C18:1 and C18:2, glutamate, ornithine, and taurine |
5.2 |
|
3 |
Lyso PC a C16:0, lyso PC a C16:1, lyso PC a C17:0, lyso PC a C18:0, lyso PC a C18:1, lyso PC a C18:2, lyso PC a C20:3, lyso PC a C20:4 |
4.7 |
- Given the complexity and length of Table 4, I suggest showing these results in a Figure, with three vertical panels, one for each pattern. Each panel can include all exposures (in the same order) and show effect estimates and 95%CI (horizontally), with discovery and validation results below each other. Exposure that belong together can be colored accordingly. Validated exposures can be marked accordingly. Table 4 can then be moved to the supplement.
Authors’ response:
Thank you for this suggestion, which we agree is well-suited for the paper. We now include a figure based on your description, which shows the effects estimates and confidence intervals in the discovery/validation sets, and have moved Table 4 to the Supplementary Materials (Now titled Table S2).
Authors’ action:
Please see page 8, Figure 1 (a, b, and c) (as shown below), and Supplementary Table 2 in Supplementary Materials (not shown below):
Figure 1. Association of dietary and lifestyle factors with metabolite patterns. Estimates presented in bold are for dietary and lifestyle factors with both significant Padj associations in the discovery dataset (blue) and p < 0.05 in the validation dataset (red). Estimates with associations that do not pass FDR threshold are presented with hollow circles.
- I suggest using left alignment for all table columns.
Authors’ response:
Thank you for your suggestion. We have center-aligned the tables according to the journal guidelines, but we are in touch with the editors to keep left alignment in the first column of Table 3 (pages 6-7), so that information is not lost.

Reviewer 2 Report
My comments mainly concern the methodology of the conducted studies and the correctness of their mutual comparison.
I would recommend extending the information on the analytical protocol used, which, according to the authors, showed high reproducibility between instruments, and individual substudies, only citation was included.
Author Response
We thank the reviewers for their detailed and helpful comments.
Below is a point-by-point response to the reviewer’s comments.
Reviewer #2:
My comments mainly concern the methodology of the conducted studies and the correctness of their mutual comparison.
- I would recommend extending the information on the analytical protocol used, which, according to the authors, showed high reproducibility between instruments, and individual substudies, only citation was included.
Authors’ response:
Thank you for this suggestion. We have included information on the methods and instruments used in the Methods section, lines 136 to 143. Regardless of sub-study, all samples were previously assayed at IARC in Lyon, France using the AbsoluteIDQ p180 Kit (Biocrates Life Sciences AG, Innsbruck, Austria), and following the procedure recommended by the vendor. To quantify metabolites, liquid chromatography mass spectrometry (LC-MS) was applied. All samples were assayed using one LC instrument (Agilent 1290, Santa Clara, CA) coupled with one of two different triple quadrupole MS instruments (Triple Quad 4500, AB Sciex, Framingham, MA for prostate and colorectal cancer and Q-Trap 5500, AB Sciex, MA for liver and kidney cancer studies.
The sources of variability in these metabolite data, including due to these sub-studies and factors relating to technical processing, have been studied in detail and previously reported on (Fages et al. Investigating sources of variability in metabolomic data in the EPIC study: the Principal Component Partial R-square (PC-PR2) method. Metabolomics 10, 1074–1083 (2014). https://doi.org/10.1007/s11306-014-0647-9). Notably, there is variation in metabolite levels by these EPIC sub-studies, which is due to a variety of factors including some variation in the instruments used, but the mixed effects models have been demonstrated to successfully correct for the unwanted variability while preserving biological variability (Jauhiainen et al. Normalization of metabolomics data with applications to correlation maps, Bioinformatics,30(15),2155–2161(2014). https://doi.org/10.1093/bioinformatics/btu175).
Please see figure below for visual representation for variation by sub-study before and after pipeline application taken from: Viallon et al. A New Pipeline for the Normalization and Pooling of Metabolomics Data. Metabolites, 11, 631 (2021). https://doi.org/10.3390/ metabo11090631.
